

# Cortisol, progesterone, 17α-hydroxyprogesterone, and TSH responses in dogs injected with low-dose lipopolysaccharide

Nicole L.B. Corder-Ramos[1,*], Bente Flatland[1], Michael M. Fry[1], Xiaocun Sun[2], Kellie Fecteau[1] and Luca Giori[1,*]

[1] Biomedical and Diagnostic Sciences Dept., University of Tennessee—College of Veterinary Medicine—Knoxville, Knoxville, TN, United States of America

[2] Office of Information and Technology, University of Tennessee—Knoxville, Knoxville, TN, United States of America

[*] These authors contributed equally to this work.

Corresponding author
Luca Giori, lgiori@utk.edu

## ABSTRACT

**Background**. Stress and diseases such as endotoxemia induce cortisol synthesis through a complex biosynthetic pathway involving intermediates (progesterone, and 17α-hydroxyprogesterone (17α-OHP)) and suppression of the hypothalamus-pituitary-thyroid axis.

**Objective**. To measure plasma concentrations of cortisol, progesterone, 17α-OHP, and thyroid stimulating hormone (TSH) in dogs experimentally injected with intravenous low-dose lipopolysaccharide (LPS). Our hypothesis was that LPS treatment would elicit a significant increase in cortisol and its precursors, and a significant decrease in TSH concentration.

**Methods**. Hormone measurements were performed on blood samples left over from a previous investigation (2011) on the effect of low-dose LPS on hematological measurands. Five sexually intact female dogs, none in estrous at the time of the study, were administered saline treatment two weeks prior to LPS treatment. LPS was administered intravenously at a dose of 0.1 µg/kg. Blood was collected before (baseline, time -24 hours) and 3-, 6- and 24-hours post-injection. Mixed model analysis for repeated measures was used, with both treatment and time as the repeated factors. Ranked transformation were applied when diagnostic analysis exhibited violation of normality and equal variance assumptions. Post hoc multiple comparisons were performed with Tukey's adjustment. Statistical significance was defined as $p < 0.05$.

**Results**. Significant differences relative to baseline values were detected following both treatments. Compared to baseline, dogs had significantly higher cortisol and 17α-OHP at 3-hours, and significantly lower TSH at 3- and 6-hours following LPS treatment. Dogs had significantly lower TSH at 6- and 24- following saline treatment. Though not statistically significant, the trend in progesterone concentrations was similar to cortisol and 17α-OHP, with an increase at 3-hours post-injection followed by a decrease close to baseline following both LPS and saline. Cortisol and 17α-OHP concentrations were higher after LPS treatment than after saline treatment at 3- and 6-hours post-injection, but differences were not statistically significant, and no significant differences between treatments were detected for any other hormone or timepoint.

**Discussion and conclusion**. Cortisol and its adrenal precursors are released in the bloodstream following a low dose of LPS, while TSH appears to decrease. Similar changes occurred following saline treatment, suggesting that even routine handling and saline injection in conditioned dogs can elicit alterations in the internal equilibrium with subsequent modification of both hypothalamus-pituitary-adrenal and thyroid axes. Changes to adrenal and thyroid hormone concentrations must be interpreted in light of clinical information. Further studies are needed to elucidate mechanisms of adrenal steroidal hormone synthesis and secretion in response to various stressful stimuli in both neutered and intact animals.

## INTRODUCTION

Endotoxin, a toxic heat-stable lipopolysaccharide (LPS) substance present in the outer membrane of gram-negative bacteria is a potent proinflammatory agent that commonly exerts pathologic, potentially life-threatening effects on humans and animals with naturally-occurring disease (*Moran, Prendergast & Appelmelk, 1996*; *Roach et al., 2005*; *Munford, 2008*). Experimental administration of LPS is used to study the pathophysiology of sepsis (*Hardaway, 2000*; *Flatland et al., 2011*; *Holowaychuk et al., 2012*; *Yu, Kim & Park, 2012*; *Lee et al., 2013*). Ample evidence supports the interconnectedness of inflammatory and hormonal pathways. Sepsis and endotoxemia elicit strong and prolonged activation of both the hypothalamus-pituitary adrenal and thyroid axes (*Kondo et al., 1997b*; *Beishuizen & Thijs, 2003*; *Kanczkowski et al., 2015*). Effects of cortisol and other adrenocorticosteroids on the immune and nervous systems have been described in humans and several animal species (*Meij et al., 1997*; *Folan et al., 2001*; *Lisurek & Bernhardt, 2004*; *Ammersbach et al., 2006*; *Feng et al., 2014*; *Kanczkowski et al., 2015*; *Dembek et al., 2017*). In vivo and in vitro evidence support a bidirectional relationship between the hypothalamus-pituitary-adrenal-axis (HPA axis) and the immune system: tissue or cellular damage stimulates macrophages and lymphocytes to release inflammatory cytokines (such as IL-1, IL-6 and TNF α) that promote hypothalamic and pituitary secretion of CRH and ACTH, respectively, which increase plasma steroid concentrations. In turn, the increased adrenal corticosteroid response has inhibitory effects on the production and action of these immune inflammatory mediators, indicating the existence of a feedback loop, where immuno-regulatory cytokines, and adrenal hormones act as afferent and efferent hormonal signals, respectively (*Meil & Mol, 2008*). Systemic inflammation causes down-regulation of thyrotropin-releasing hormone (TRH), leading to lowered secretion of thyroid stimulating hormone (TSH), total T4 and T3 (*Yu, Kemppainen & MacDonald, 1998*; *Straub, 2014*). Increased cortisol from activation of the HPA axis can also impact thyroid function and thyroid hormone metabolism, affecting both the hypothalamic and pituitary release of stimulating factors and decreasing the deiodination of thyroxine (T4) into the metabolically active triiodothyronine

(T3) in peripheral tissues (*Peterson et al., 1984*; *Ferguson & Peterson, 1992*; *Meij et al., 1997*; *Daminet & Ferguson, 2003*).

To the authors' knowledge, there have been no previous investigations of the effects of experimental LPS administration on plasma adrenocorticosteroids or TSH concentrations in dogs. The objective of the present study was to investigate the effects of intravenous (IV) low-dose (0.1 μg/kg) LPS administration on plasma concentration of cortisol, its precursors (progesterone, 17α-OHP), and TSH, in dogs. We hypothesized that low-dose LPS administration would result in increased concentration of plasma cortisol, progesterone, and 17α-OHP, and decreased plasma concentration of TSH.

## MATERIALS AND METHODS

### Study design and sample collection and banking

This study used samples remaining from an earlier investigation in canines administered low-dose LPS (*Flatland et al., 2011*). Study design and methodology are described in detail elsewhere (*Flatland et al., 2011*). Briefly, five sexually intact female dogs, none in estrous at the time of the study, were injected intravenously with physiologic saline and, after a two-week washout period, with LPS (0.1 μg/kg IV). All dogs received subcutaneous crystalloid fluids (20 mL/kg) prior to treatment (saline and LPS). Blood samples were collected into EDTA tubes 24 h before (baseline) and at 3-, 6- and 24-hours after both treatments. Hematological analyses were performed within 30 min of blood collection. The remaining blood from each sample was centrifuged at $700\times$ g for 10 min (Sero-Fuge centrifuge; Becton, Dickinson and Co, Franklin Lakes, NJ, USA), and plasma was separated and stored in sterile microcentrifuge tubes at $-80\,°C$ until use in 2017.

### Hormone measurements

Hormone assays were performed in the Clinical Endocrinology Laboratory at the University of Tennessee College of Veterinary Medicine. Frozen plasma from the earlier study was thawed by removing specimens from the $-80\,°C$ freezer and allowing them to equilibrate with room temperature. Cortisol, progesterone, and TSH were measured using a chemiluminescence immunoassay system (Immulite 1000; Siemens Healthcare Diagnostics Products Ltd., Los Angeles, CA, USA) and specific reagent kits for each hormone (Cortisol: LKCO1 kit; Progesterone: LKPG kit; TSH: Canine TK9 kit; Siemens Healthcare Diagnostics Products Ltd., Los Angeles, CA, USA). 17α-OHP measurement was performed by radioimmunoassay (RIA) using commercial reagent antibody (ImmuChem[TM] Double Antibody 17-Hydroxyprogesterone kit; MP Biomedicals, LLC, Orangeburg, NY, USA). The radioactivity of the supernatant was measured using a gamma counter (Packard Cobra II Auto-Gamma, Packard Instrument Company, Meridian, CT, USA)). Cortisol, progesterone, and TSH concentrations were measured once (Immulite) while 17α-OHP concentrations were measured in duplicate (RIA). The Immulite analyzer routinely performs 12 replicate measurements for each sample and reports the results as mean of the ten readings, after excluding the lowest and highest values. Hormone analyses were performed on completely thawed samples, all in one day. All hormone measurements were
performed in accordance with the manufacturer's instructions by personnel trained in use of the instruments and test kits.

## Data analysis

The effects of treatment and time on response variables (cortisol, progesterone, 17α-OHP, and TSH) were analyzed respectively using mixed model analysis for repeated measures with the experiment unit ID as the random effect while both treatment and time as the fixed factors. Mixed model analysis was adopted in current analysis because both random effect (subject) and fixed effects (treatment and time) are involved in the experimental design. Ranked transformation were applied on each response variable because diagnostic analysis for Shapiro–Wilk test and Levene's test on residuals exhibited violation of normality and equal variance assumptions. Post hoc multiple comparisons were performed with Tukey's adjustment. Statistical significance was identified at the level of 0.05. All analyses were conducted in SAS 9.4 TS1M4 (SAS institute Inc., Cary, NC, USA). Data were presented as mean ± standard error. Statistical comparisons made were results at 3, 6, and 24 h, respectively, vs. baseline following each treatment (saline vs. LPS). Results following each treatment at each timepoint were also compared to each other (baseline saline vs. baseline LPS, 3- hour saline vs. 3-hour LPS, etc.). Data at each timepoint included results from all 5 dogs, except for 24-hours. At 24 h, results included data from only 4 dogs, because one sample from dog 2 (after saline treatment) and one sample from dog 3 (after LPS treatment) were unavailable for testing.

## RESULTS

### Cortisol

Raw data are given in Table 1 and depicted in Fig. 1. Mean for cortisol concentrations in all animals following both treatments at each time point are listed in Table 2. Following LPS treatment, plasma cortisol concentration was significantly increased at 3 h post injection versus baseline. No significant differences were observed at 6- or 24-hours versus the baseline concentration. Following saline treatment, significant difference was not observed for any time points versus baseline. When data for each treatment were compared, a significant difference was detected only at 6-hours post-injection. No significant differences for LPS vs. saline were observed at any other timepoint (baseline, 3- and 24-hours).

### Progesterone

Raw data are given in Table 1 and depicted in Fig. 2. Mean progesterone concentrations in all animals following both treatments at each time point are listed in Table 2. Progesterone concentrations from dog 1 were excluded from analysis because these were above institutional reference intervals calculated using data from a group of healthy intact female in anestrus. Although changes were observed, particularly at 3-hours post-injection, no changes were statistically significant between treatments at the same time points nor between any time point and baseline concentrations for each treatment.

**Table 1** Hormone concentrations in five dogs treated IV with saline or low-dose LPS.

| | | SALINE | | | | LPS | | | |
|---|---|---|---|---|---|---|---|---|---|
| | Collection time (hours) | Cortisol μg/dL | Progesterone ng/mL | 17α-OHP ng/mL | TSH ng/mL | Cortisol μg/dL | Progesterone ng/mL | 17α-OHP ng/mL | TSH ng/mL |
| Dog 1 | baseline | 2.9 | 36.7 | 3.31 | 0.14 | 3.5 | 25.4 | 3.01 | 0.12 |
| | 3 | 1.7 | 32.5 | 3.42 | 0.12 | 17.0 | 15.2 | 9.57 | 0.04 |
| | 6 | 1.2 | 29.4 | 2.99 | 0.1 | 4.3 | 10.5 | 3.56 | <0.03 |
| | 24 | 1.1 | 31.6 | 2.75 | QNS | 1.9 | 13.4 | 2.23 | 0.07 |
| Dog 2 | baseline | 1.2 | <0.20 | 0.08 | 0.21 | 1.3 | 0.52 | 0.22 | 0.07 |
| | 3 | 8.6 | 0.33 | 0.46 | 0.13 | 16.5 | 3.49 | 3.69 | <0.03 |
| | 6 | 1.0 | <0.20 | 0.12 | 0.04 | 2.7 | 0.68 | 0.65 | <0.03 |
| | 24 | NA | NA | NA | NA | 1.2 | 0.39 | 0.42 | <0.03 |
| Dog 3 | baseline | 1.2 | 3.57 | 0.86 | 0.07 | 1.6 | <0.20 | <0.08 | 0.14 |
| | 3 | 15.9 | 6.35 | 4.7 | 0.04 | 16.0 | 1.33 | 1.46 | 0.04 |
| | 6 | 1.3 | 3.49 | 1.06 | <0.03 | 11.4 | 0.65 | 0.65 | 0.06 |
| | 24 | 1.4 | 3.13 | 1.56 | <0.03 | NA | NA | NA | NA |
| Dog 4 | baseline | 2.2 | 0.95 | 0.29 | 0.09 | 1.4 | 0.58 | 0.19 | 0.12 |
| | 3 | 25.5 | 6.08 | 4.33 | 0.03 | 25.8 | 4.99 | 5.5 | <0.03 |
| | 6 | 3.4 | 1.24 | 0.65 | <0.03 | 5.5 | 0.67 | 0.9 | <0.03 |
| | 24 | 1.4 | 0.82 | 0.68 | 0.04 | 1.9 | 0.66 | 0.58 | 0.04 |
| Dog 5 | baseline | 1.5 | <0.20 | 0.91 | 0.18 | 3.1 | <0.20 | 1.13 | 0.15 |
| | 3 | 1.9 | <0.20 | 1.41 | 0.08 | 18.7 | 4.69 | 8.56 | 0.05 |
| | 6 | 2.2 | <0.20 | 0.97 | 0.07 | 13.2 | 2.13 | 5.4 | 0.03 |
| | 24 | <1.0 | <0.20 | 0.66 | 0.05 | 4.1 | <0.20 | 1.31 | 0.04 |

**Notes.**
QNS, Quantity not sufficient; NA, Not applicable.
Dark gray-shaded represent excluded data.

### 17α-OHP

Raw data are given in Table 1 and depicted in Fig. 3. Mean for 17α-OHP concentrations in all animals following both treatments at each time point are listed in Table 2. Similar to cortisol, following LPS treatment, plasma 17α-OHP concentration was significantly increased only at 3 h post injection versus baseline. No significant differences were observed at 6- and 24-hours versus baseline concentration. Following saline treatment, significant difference was not observed for any time points versus baseline. No significant differences were observed for LPS vs. saline at the same time points (baseline, 3-, 6- and 24-hours).

### TSH

Raw data are given in Table 1 and depicted in Fig. 4. Mean for TSH concentrations in all animals following both treatments at each time point are listed in Table 2. After LPS treatment, plasma TSH concentration was significantly decreased both at 3- and 6-hours post injection versus baseline. No significant differences were observed at 24-hours versus baseline concentration. After saline treatment, TSH concentration was significantly decreased both at 6- and 24-hours post injection versus baseline. No significant differences

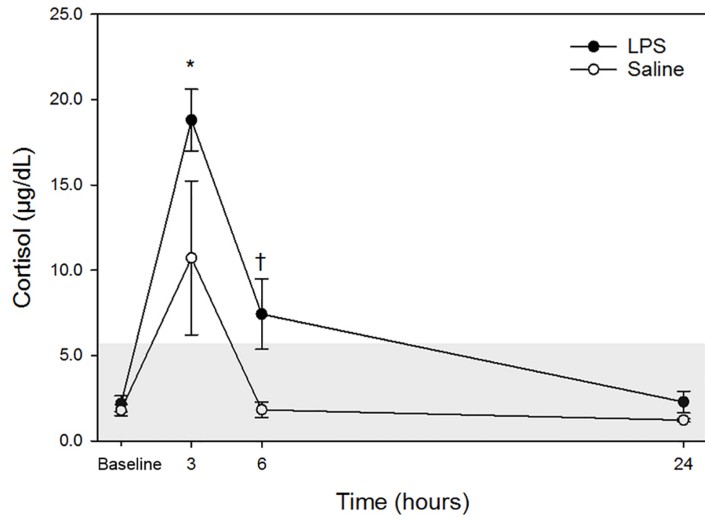

**Figure 1    Mean plasma cortisol concentrations in five dogs pre-treatment (baseline), at 3-, 6- and 24-hours post-injection with saline and LPS.** The institutional reference interval for canine cortisol (based on $n = 95$ clinically healthy dogs of varying breeds) is represented by the shaded grey area from $< 1.0$–$6.0$ µg/dL. †$P < 0.05$ for between group differences; *$P < 0.05$ versus time 0 within LPS group.

were observed at 3-hours versus baseline concentration. No significant differences were observed for LPS vs. saline at the same time points (baseline, 3-, 6- and 24-hours).

## DISCUSSION

LPS is an important inflammatory stimulus. The LPS molecule itself is capable of inducing a septic response upon sufficient exposure (*Munford, 2016*). Restoration of internal equilibrium after endotoxemia occurs through many mechanisms, including activation of the HPA axis by neuroendocrine and immune mechanisms (*Cavaillon, 1990*; *Elenkov et al., 1992*; *Grinevich et al., 2001*; *Webster, 2004*). Briefly, endotoxemia induces an acute phase response that involves activation of signaling cascades leading to release of pro-inflammatory cytokines, such as interleukin-1 (IL-1), interleukin-6 (IL-6), and tumor necrosis factor- α (TNF- α) (*Perlstein et al., 1993*; *Tracey, 2002*; *Beishuizen & Thijs, 2003*). Circulating cytokines affect neuronal signaling in the HPA axis, resulting in the production of glucocorticoids, which in turn produce anti-inflammatory effects that restore equilibrium (*Sternberg, 2006*).

In this investigation, cortisol and progestin (progesterone and 17 α-OHP) concentrations were above institutional reference intervals at 3-hours post-injection after both saline LPS treatments. Production of cortisol from cholesterol occurs through a complex biosynthetic pathway involving various precursors, including intermediates such as progesterone, and 17α-hydroxyprogesterone (17α-OHP) which are released in the bloodstream together with the main glucocorticoid (*Lisurek & Bernhardt, 2004*; *Mullington, 2009*). Therefore, a significant increase in progesterone and 17α-OHP at 3-hours was expected following LPS treatment. However, a boost in cortisol, progesterone, and 17α-OHP concentrations was also observed following saline treatment, suggesting that restraint and/or saline injection

**Table 2** Descriptive statistic for all hormones at each time point in all dogs from saline and LPS groups.

| Measure | Treatment | n obs. | Time | Mean |
|---|---|---|---|---|
| Cortisol | LPS | 5 | baseline | 2.18 |
| | | 5 | 3 | 18.80[*] |
| | | 5 | 6 | 7.42[†] |
| | | 4 | 24 | 2.28 |
| | SALINE | 5 | baseline | 1.80 |
| | | 5 | 3 | 10.72 |
| | | 5 | 6 | 1.82 |
| | | 4 | 24 | 1.23 |
| Progesterone | LPS | 4 | baseline | 0.38 |
| | | 4 | 3 | 3.63 |
| | | 4 | 6 | 1.03 |
| | | 3 | 24 | 0.42 |
| | SALINE | 4 | baseline | 1.23 |
| | | 4 | 3 | 3.24 |
| | | 4 | 6 | 1.28 |
| | | 3 | 24 | 1.38 |
| 17αOHP | LPS | 5 | baseline | 0.93 |
| | | 5 | 3 | 5.76[*] |
| | | 5 | 6 | 2.23 |
| | | 4 | 24 | 1.14 |
| | SALINE | 5 | baseline | 1.09 |
| | | 5 | 3 | 2.86 |
| | | 5 | 6 | 1.16 |
| | | 4 | 24 | 1.41 |
| TSH | LPS | 5 | baseline | 0.12 |
| | | 5 | 3 | 0.04[*] |
| | | 5 | 6 | 0.04[*] |
| | | 4 | 24 | 0.05 |
| | SALINE | 5 | baseline | 0.14 |
| | | 5 | 3 | 0.08 |
| | | 5 | 6 | 0.05[*] |
| | | 3 | 24 | 0.04[*] |

**Notes.**
[*]$P < 0.05$ versus time 0 within each group.
[†]$P < 0.5$ for between group differences.

elicited a cortisol based physiologic stress response in some dogs. Two of the five dogs did not show any changes compatible with activation of the HPA axis after injection of saline at any time point, while the other three showed elevations of cortisol and progestins ranging between 7- and 12-fold above baseline concentrations, mimicking, in at least two animals (dogs 3 and 4) their responses to LPS. These dogs were part of a research kennel and were conditioned to being handled for research projects. Nevertheless, these results suggest that a simple manipulation pre-experiment or the injection of a placebo

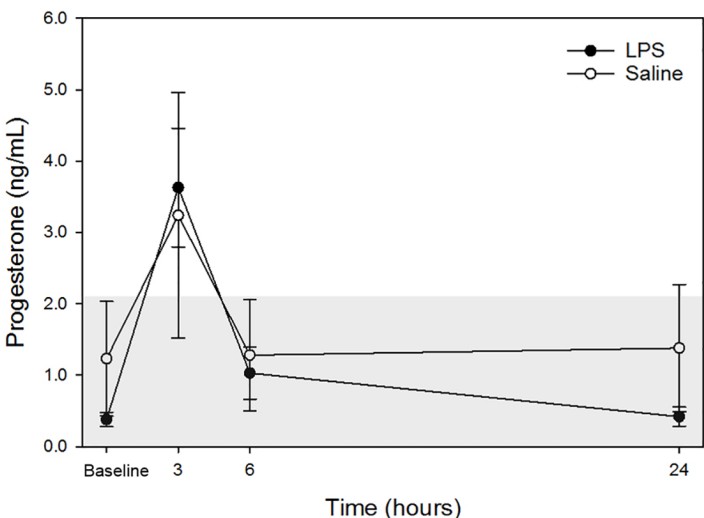

**Figure 2** **Mean plasma progesterone concentrations from 4 dogs pre-treatment (baseline), at 3-, 6-, and 24-hours post-injection with saline and LPS.** Data from one dog suspected to be in diestrus were excluded (see discussion for detail). The institutional reference interval for canine progesterone (based on $n = 20$ clinically healthy intact female in anestrus of varying breeds) is represented by the shaded grey area from $< 0.20 - 2.16$ ng/mL.

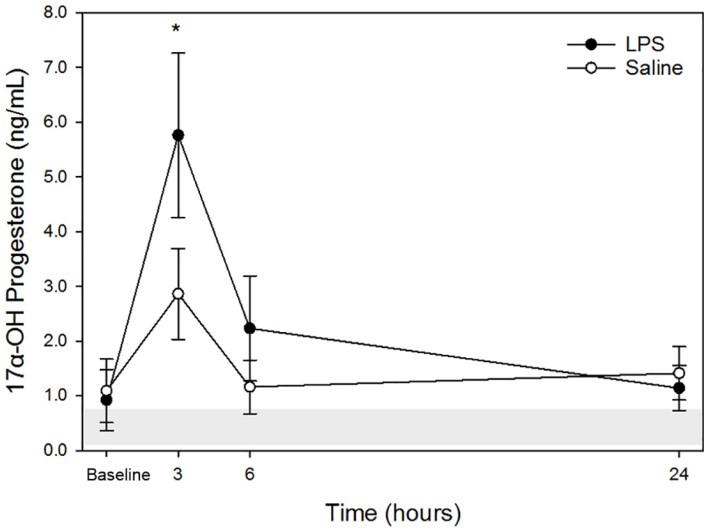

**Figure 3** **Mean plasma 17 $\alpha$-OHP concentrations from five dogs pre-treatment (baseline), at 3-, 6-, and 24-hours post-injection with saline and LPS.** The institutional reference interval for canine 17 $\alpha$-OHP (based on $n = 20$ clinically healthy intact female in anestrus of varying breeds) is represented by the shaded grey area from 0.08–0.69 ng/mL. *$P < 0.05$ versus time 0 within LPS group.

substance can perturb body homeostasis and activate the HPA axis in some individuals. Implications are that apparently theoretically non-stressful events such as routine handling and placebo injections can impact physiology, an important consideration when studying adrenal hormones and stress. As for clinical settings, changes to adrenal hormones must be

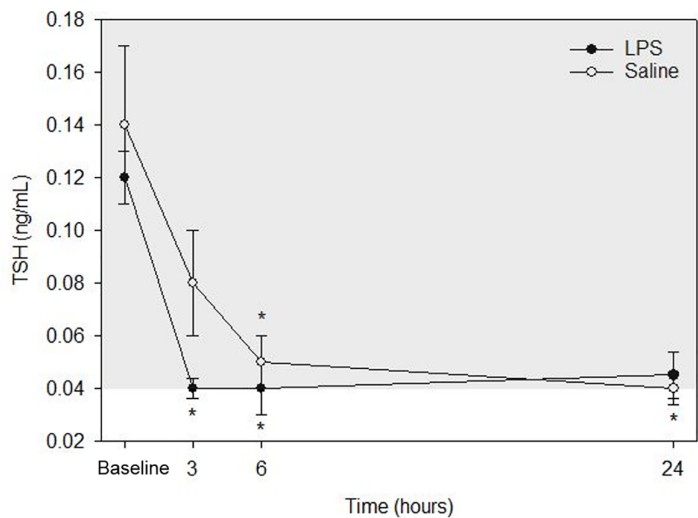

**Figure 4 Mean plasma TSH concentrations from five dogs pre-treatment (baseline), at 3-, 6-, and 24-hours post-injection with saline and LPS.** The institutional reference interval for canine TSH (based on $n = 50$ clinically healthy dogs of varying breeds) is represented by the shaded grey area from 0.04–0.38 ng/mL. *$P < 0.05$ versus time 0 within each group.

interpreted in light of anamnesis, clinical signs, other clinico-pathological abnormalities, previous diagnostic tests, and prevalence of disease (i.e., for infectious diseases) (*Akobeng, 2007*; *Sikkens et al., 2016*).

The observed inverse relationship between cortisol and TSH concentrations is expected and can be explained by the fact that LPS affects the HPT axis by decreasing thyroid and thyrotropin stimulating hormones, and downregulating thyroid receptors on target tissues (*Van der Poll et al., 1999*; *Beigneux et al., 2003*). Cortisol has been shown to exhibit negative feedback on the HPT axis at the level of the hypothalamus, producing a significant decrease in TSH (*Nicoloff, Fisher & Appleman, 1970*). In our study, a 70% decrease from baseline concentration was observed in TSH 3-hours post-injection with LPS, in association with the peak increase in cortisol at the same time point. TSH concentrations were close to the lower reference limit both at 6- and 24-hours after injection. A similar trend was observed following saline administration, with a significant decrease in TSH at 6- and 24-hours post-injection. Although decreased TSH concentration following LPS administration is most likely due to the aforementioned endotoxin effect on the hypothalamus and thyroxin receptors, findings after saline administration suggest that cortisol and progestins alone could affect TSH synthesis and illustrate that thyroid hormones are exquisitely sensitive to physiologic stress.

It is well documented that glucocorticoids impact leukocyte kinetics. In dogs, a corticosteroid-mediated leukogram ("stress leukogram") is characterized by mature neutrophilia, lymphopenia, monocytosis, and eosinopenia (*Bertók, 1998*; *Weiss, Wardrop & Schalm, 2010*; *Petrie, 2010*). Cortisol increases neutrophil release from the bone marrow storage pool and downregulates neutrophil L-selectin expression, decreasing adhesion to endothelial cells and allowing neutrophil migration across the vascular wall (*Burton*

*et al., 1995*, p. 1; *Berton et al., 1996*; *Miles et al., 1998*; *Radi, Kehrli & Ackermann, 2001*). L-selectin downregulation shifts neutrophils from the marginated pool to the circulating pool, causing an increase in neutrophil concentration as measured in blood (*Stockham & Scott, 2008*). Multiple mechanisms has been suggested and documented to explain stress-induced lymphopenia, which include both decreased efflux from lymph nodes, and decreased proliferative and activation cytokines (e.g., IL-2) for lymphocytes (*Stockham & Scott, 2008*). Hematologic changes documented in the study that preceded the present one were consistent with cortisol-mediated effects in saline-treated dogs—effects that were overshadowed by more dramatic direct effects of endotoxemia in LPS-treated dogs (*Flatland et al., 2011*).

This study has some limitations. The dogs were all sexually intact, adult females not in estrus at the time of the experiment. Based on hormone analysis, all dogs were considered in anestrus except one (dog 1) that had high progesterone and 17α-OHP concentrations during the entire 2 weeks of the experiment (at all-time points after saline or LPS injections) (see dog 1 in Table 1). Additionally, another dog (dog 3) had mildly increased progesterone concentrations in the first part of the experiment (at all-time points following saline). Dog 1 was most likely in diestrus during the entire study, with increased progesterone concentrations due to a persistent corpus luteum. It seems likely that a large amount of progesterone originating from the corpus luteum masked subtler changes in progesterone concentration originating from the adrenal glands. Although 17α-OHP increases were of a similar magnitude as progesterone changes, concentrations of this hormone were lower and changed at different time points following LPS treatment, suggesting that 17α-OHP was released by the adrenal glands rather than a corpus luteum. It is known, that 17α-OHP is primarily produced in the adrenal glands and only to some degree in the corpus luteum and gonads (*Honour, 2014*; *Karagüzel et al., 2019*). In contrast to dog 1, progesterone concentrations in dog 3 were mildly above institutional reference intervals only following saline treatment at each time point. Dog 3 was most likely in an advanced diestrus phase during the first (saline treatment) part of the study and in anestrous during the second part (LPS treatment). 17α-OHP concentrations in dog 3 were just above the reference intervals only after saline administration, suggesting that a corpus luteum could have contributed to mildly increased 17α-OHP in this dog during this study phase.

A further limitation is that the study design did not control for day (i.e., saline treatments done first in all dogs, followed by LPS treatments). Husbandry, dog handling, and sampling conditions were kept constant throughout the experiment. Nonetheless, we cannot exclude an effect of time (due to treatment sequence, innate biological variation and/or undetected changes in the dogs' environment or health status) on results. Despite this limitation, data from this study and the one that preceded it support a true effect of LPS on plasma cortisol, 17α-OHP, and TSH concentrations.

Finally, plasma samples used for this study had been stored at −80 °C for 6 years without undergoing any thaw cycles. Use of stored samples may be seen as a limitation; however, numerous publications attesting to the high stability of steroid and thyroid hormones when preserved at −80 °C have been published (*Kubasik et al., 1982*; *Kley, Schlaghecke &*

*Krüskemper, 1985*; *Garde & Hansen, 2005*; *EL Ezzi, El-Saidi & Kuddus, 2010*; *Hillebrand, Heijboer & Endert, 2017*).

## CONCLUSIONS

Cortisol, progesterone, and 17α-OHP exhibited similar trends in concentration at all timepoints following treatment with saline or LPS. The increases in adrenal steroid precursors and cortisol following LPS treatment supported our hypothesis. The data also showed that the concentration of those hormones increased, albeit less consistently, after saline treatment—suggesting that even a placebo treatment in conditioned subjects can elicit activation of the HPA axis. Similarly, the decrease in TSH following LPS treatment supported our hypothesis, and a milder decrease also occurred after saline treatment— again, suggesting that even placebo treatment can affect thyroid hormone pathways. Changes to adrenal and thyroid hormones must be interpreted in light of anamnesis, clinical signs, and other clinico-pathological abnormalities.

This is an observational study that did not investigate pathogenetic mechanisms. Changes reported here followed one particular type of inflammatory or placebo stimulus, and mechanisms and physiologic responses may or may not be similar across other stimuli. Further studies would be needed to elucidate mechanisms of adrenal steroidal hormone synthesis and secretion in response to various stimuli in both neutered and intact animals.

**Abbreviations and Symbols**

| | |
|---|---|
| **17α-OHP** | 17α-hydroxyprogesterone |
| **ACTH** | Adrenocorticotropic hormone |
| **ANOVA** | Analysis of variance |
| **CRH** | Corticotropin-releasing hormone |
| **HPA** | Hypothalamic-pituitary-adrenal |
| **HPT** | Hypothalamic-pituitary-thyroid |
| **IL-1** | Interleukin-1 |
| **IL-6** | Interleukin-6 |
| **LPS** | Lipopolysaccharide |
| **RIA** | Radioimmunoassay |
| **TNF-α** | Tumor necrosis factor-α |
| **TSH** | Thyroid stimulating hormone |
| **WBC** | White blood cell |

## ACKNOWLEDGEMENTS

The authors thank Julie Fields and Joseph Ramos from the Department of Biomedical and Diagnostic Sciences at the University of Tennessee College of Veterinary Medicine for technical assistance.

### Funding

The authors received no funding for this work.

### Competing Interests

The authors declare there are no competing interests.

### Author Contributions

- Nicole L.B. Corder-Ramos analyzed the data, prepared figures and/or tables, authored or reviewed drafts of the paper.
- Bente Flatland performed the experiments, authored or reviewed drafts of the paper, approved the final draft.
- Michael M. Fry performed the experiments, approved the final draft, critically revised the article.
- Xiaocun Sun analyzed the data, statistical analysis.
- Kellie Fecteau contributed reagents/materials/analysis tools, critically revised the article.
- Luca Giori conceived and designed the experiments, analyzed the data, contributed reagents/materials/analysis tools, prepared figures and/or tables, authored or reviewed drafts of the paper, approved the final draft.

### Data Availability

The raw measurements are available in Table 1.

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
