# Peer review of "Cortisol, progesterone, 17α-hydroxyprogesterone, and TSH responses in dogs injected with low-dose lipopolysaccharide"

_PeerJ, doi:10.7717/peerj.7468_

## Round 0.1 · original submission · Minor Revisions

The authors need to better address statistical and technical issues, as pointed out by the reviewers.

·

Basic reporting

- Everywhere where you say “saline and LPS”, I would replace by “saline or LPS” for improved clarity
- You can’t afford to say “various time points” within the material and method part: this is the only part where everything needs to be detailed: this is the part where the reader goes for clarification or precision in the study design
- List the Siemens kits for each hormone other than 17OHP4

Experimental design

The study design is unclear, the following points need to be specified:
- Emphasize more that there were 5 dogs total and they got both treatments in a row in the same order (rather than twice 5 different dogs, meaning 10 dogs total – I understood it only in the discussion once explaining the limits). This needs to be addressed in details within the material and method.
- Why did the dogs get crystalloids before each treatment? And how long before each treatment? And when compared to the pre-injection blood drawing for hormones measurements?
- If baseline corresponds with 24 hours before treatment injection, then clarify, as the notation “baseline, time -24 hours” seems unclear to me – one wonders if the two propositions separated by a coma are the same or different.
- You also need to specify the assumed ovarian cycle status of the bitches within the material and method, the reader can’t discover this in the discussion. I would add something like: “bitches were selected to be for sure outside of the estrus phase of the ovarian cycle, and hopefully in anestrus.”
- Please add one sentence to explain what a mixed model analysis is. Were the results analyzed by ANOVAs or not?
- It is unclear to me which statistical method you used to come up with the results:
o You need to specify what was analyzed for normality and equal variance assumption; just saying “analysis” is insufficient to allow the reader to picture what you did. Was it the set of 4 results for each dog for each time point for each hormone? If so, how reliable is a normality test performed on only 4 values, especially with a cutoff for the p value at 0.05?? Probably extremely low.
o Which normality test did you use?
o It is because you mentioned checking normality and equality of variances that I deducted that you used ANOVAs, but this is not mentioned anywhere actually; was it actually the case? If yes, then specifies that you used Tuckey to address the result of the ANOVAs. If not, then you really need to re-explain your statistical approach: if the reader can’t picture what you did, it’s difficult to trust the result.
o (Did you use Tuckey or another test to address the results that did not verify normality and equality of variances?)
- I would mention the delay of the freezing (2011 – 2017) in the Summary

Are you sure that the Immulite does indeed 10 measurements and provides the mean? I had never heard this and tend to doubt it; I thought it measured the value only once (whereas RIA does indeed 2 measurements and provides a mean with a CV). The Immulite uses an antibody-coated bead for each measurement, and I’m almost sure it needs one bead per measurement. I don’t think it uses 10 beads per value to perform 1 measurement: to be double-checked.

Validity of the findings

Cortisol results:
- Are you sure the saline was not significantly increased at 3 hours versus baseline ?
TSH:
- Are you sure there was no significant difference between 24 hours and baseline ?? The figure really shows the opposite…

To me, 2 findings need to be added and discussed:
1) Limitations about the normality tests (and thus all the following stats) for tests about 4 values
(+the chosen normality test needs to be all mentioned, as all other statistical tests)
2) It seems very important to me that:
a. All steroid hormones increased clearly after LPS in the 5 dogs
b. Whereas no steroid hormones increased in 2 out of 5 dogs, meaning almost half the dogs.

The second point was mentioned, but too late in the article in my opinion, and it probably needs more discussion, especially mentioning that then:
- LPS triggers the steroids
- Whereas it’s probably not the saline but the process that triggers the steroids in the saline part (otherwise the steroids would increase in the 5 dogs post saline all the same)

Additional comments

Very interesting work, English is good, references are good, and the topic is definitely relevant and useful. You just need to:
- Develop the material and method
- Include a more detailed stat paragraph (mandatory in my opinion)
- And add a couple specifications mentioned above.

Reviewer 2 ·

Basic reporting

Suggestions:
Line 29- Add space between '6-' and 'and'
Line 71- Consider adding reference Holowaychuk MK, Birkenheuer AJ, Li J, Marr H, Boll A, Nordone SK. J Vet Intern Med. 2012 Mar-Apr;26(2):244-51.
Line 77- Indicate which species have been studied that are being cited.
Line 77- References Dembek eat al, Fend et al, Folan et al are not found in the references section. Please check all references and insure that they are in the references section.
Line 87- Consider addressing the elevation in FT4 seen in cats with chronic illness.
Lines 150 and 167- expand 'significance' to significant difference
Line 198- Specify cortisol based physiologic stress response (in contrast to epinephrine/norepinephrine).
Line 244- change to (at all time points after saline and LPS injection) for consistency with line 246.

Experimental design

Research objectives are clearly defined, relevant and meaningful. The findings both fill knowledge gaps as well as identify new gaps needed to investigate further.
Authors adequately address the limitations of the study as well as any concern regarding the stability of the hormones.

Validity of the findings

The findings are clearly presented and not overstated.

---

## Round 0.2 · accepted · Accept

The authors have satisfactorily addressed the issues raised by the reviewers. The further suggestions by Dr. Korchia could considered prior to publication in the galley proofs.

·

Basic reporting

Congratulation for this article as well as for all the modifications you did to address the reviewing. I wish to mention few last remarks:

1) I would not use the term “the experiment unit ID” as it makes the sentence truly confusing; if you refer to dogs, then say dogs: the reader will understand much better.
2) You provided a lot of statistical information to me that were really an excellent resource (see both bellow), and I do highly encourage you to include them into your paper (even if you modify slightly the style to fit the one for a published article), as they are of great value! I hope you will do so.
• “This mixed model analysis can be called a split-plot ANOVA, it is an ANOVA analysis not the most commonly so-called ANOVA where only fixed effects are involved”
• “Mixed model analysis was mentioned in the statistical analysis section and details regarding how the data were analyzed were summarized in this section. This analysis is not a common ANOVA, which only involves fixed effects. Therefore mixed model analysis instead of ANOVA is usually used to emphasize the experimental design. The data was analyzed by following the routine steps as follows: The raw data (not transformed) were analyzed using mixed model analysis as described in the statistical analysis section. The residuals generated were then used for conducting the diagnostic analysis for normality and equal variance assumptions. Because the model assumptions were violated (significant Shapiro-Wilk test and Levene's test), rank data transformation was conducted on the raw data. The mixed model analysis was then conducted again on rank-transformed data. Analysis of rank transformed data is a type of non-parametric analysis which assumes no normality and equal variance model assumptions and this is a common strategy to handle data with model assumption violation issues. Data was not rank transformed in the first place because the biggest drawback for all non-parametric analysis is lesser power of the analysis. Rank transformation is only applied when violation of model assumptions is verified. Mixed model analysis only reports the significance of overall effect of each factor but not the difference among each levels of each factor. Therefore, in the statistical analysis section, post-hoc multiple comparisons analysis with Tukey's adjustment was described. The multiple comparisons are post-hoc testing after the mixed model analysis, and they were also conducted on ranked data and, therefore, no model assumptions needed to be verified.”
3) I know where the confusion about the measurements of the Immulite comes from: you said “The Immulite analyzer routinely performs 120 replicate measurements”; the confusion comes from the term “replicate,” which is typically used in quality to signify a new entire test (i.e., 20 replicates for short term precision). Here, this is not a replicate, this is a series of 12 measurements (of the same replicate, if I can say so). I highly encourage to remove the term replicate to eliminate confusion from that paragraph.

Thanks, and again congratulation!

Experimental design

no comment

Validity of the findings

no comment

Additional comments

no comment